# Exploring Urban (Living) Labs: A Model Tailored for Central and Eastern Europe's Context

Bartosz Piziak * , Magdalena Bień, Wojciech Jarczewski and Katarzyna Ner

Institute of Urban and Regional Development, Cieszynska 2, 30-015 Kraków, Poland; mbien@irmir.pl (M.B.); wjarczewski@irmir.pl (W.J.); kner@irmir.pl (K.N.)
* Correspondence: bpiziak@irmir.pl

**Abstract:** The article attempts to synthesise existing knowledge and research related to the functioning of urban (living) labs and to analyse the particular experiences of their dozens of representatives from all over the world in order to develop a definition and model of an urban lab adapted to the conditions of Central and Eastern European countries. The lack of a systematised definition concerning functioning urban labs has influenced the development of a single, possibly precise definition of an urban lab, adapted to the socio-economic conditions of CEE countries. On the basis of a systematic review of the literature on the subject and a questionnaire survey of 24 urban labs from different countries of the world regarding their functioning, an attempt was made to develop an integrated model of an urban lab, taking into account elements such as stakeholder groups, thematic areas of activities, or stages of the design process, among others. The various definitions and typologies of urban (living) labs presented in this article, as well as the different approaches to their operation in many countries, indicate what an elaborate and heterogeneous tool they are. Despite the noticeable differences, their overarching goal of operation is invariably to improve the quality of life of city dwellers, taking into account the interests of different audiences. The growing interest in urban labs is reflected in the increasing number of publications on the subject of their functioning and the rising number of "urban lab" initiatives, which influences the larger number of cities considering their implementation. So far, this tool has not been used in Central and Eastern European countries, including Poland, which led the authors of this study to develop the concept of an urban lab (2018/2019), based on which a pilot project was implemented in two Polish cities between 2019 and 2021.

**Keywords:** urban lab; residents' participation; city innovations; community engagement



## 1. Introduction

The literature on the subject points to various aspects of combining urban development with the creation of innovative projects, including processes in which citizens are actively involved, e.g., hackathons, innovation jams, or urban labs [1]. The latter have become a kind of phenomenon in contemporary cities, and their development has intensified in the last decade [2]. They provide a testing ground for urban innovation and enable the promotion of new, collaborative, transdisciplinary ways of thinking in urban development planning [3]. However, as indicated in the work by K. Steen and E. van Bueren [2], it is still not entirely clear what such urban labs actually are.

The aim of this article is to synthesise the existing knowledge and research related to the functioning of urban labs in the world, in particular to analyse the definitions of urban (living) labs functioning in the literature and the forms of activities of urban laboratories from all over the world. During the search for information on urban labs worldwide as well as in the survey carried out, the following aspects were taken into account: the objectives of urban labs, the activities undertaken and their scope, the forms of cooperation with various urban stakeholders, or the priority thematic areas. On the basis of the results of these

studies and the analysis of the literature on the subject, an attempt was made to develop a single, possibly precise definition of an urban lab and its integrated functional model adapted to the socio-economic conditions of Polish cities, as this tool has neither been used so far in Poland nor in the countries of Central and Eastern Europe, which turned out to be the case in the course of the conducted research (especially taking into consideration the post-communist countries, such as the Czech Republic, Slovakia, and Hungary).

Based on the concept developed in 2018–2019, a pilot project for the implementation of urban labs in two Polish cities (Gdynia and Rzeszów) was realised, the main objectives of which were to change the approach of the inhabitants of these cities to projects conducted in cooperation with various urban stakeholders in accordance with the concept of the quadruple helix, as well as to activate the inhabitants to co-manage their cities and take action to develop innovative urban solutions.

## 2. Materials and Methods

For the purpose of this paper, a systematic literature review was carried out. Items were searched in ResearchGate, Google Scholar, and Infona databases based on keywords such as urban living lab, urban lab, living lab, and city lab. In addition to secondary research, a survey to gather basic and detailed information on the operation of urban labs around the world was also conducted. The survey was sent out by e-mail to 39 identified and selected urban labs operating in various countries worldwide, and the survey form itself was made available to interested parties online. Responses to the extended survey form were submitted between November 2018 and April 2019. The identification of the target group of the study, to whom the request to complete the questionnaire had been sent, was made through an online search of urban labs worldwide. The selection of the target group was based on the occurrence of the phrase "urban lab" or "urban living lab" in the name or description of the lab, the specificity of the activities, the level of involvement of the inhabitants and other urban stakeholders, as well as sustainability. In some cases, it was necessary to verify the data obtained from the Internet by telephone, so the response process was prolonged, and the number of urban laboratories identified was reduced, either due to the suspension of the activities of some of them, the end of the project for which they were funded, or a complete lack of contact. The final number of urban laboratories that responded to our request was 25, and for one of them, we received a request for complete confidentiality and not publishing the information provided to us in our study. Therefore, 24 completed questionnaires (17 countries) were accepted for analysis, representing 61.5% of all labs invited to the study (Appendix A).

## 3. Results

### 3.1. Theory: The Origin of the Idea and the Definition of the Urban (Living) Lab

When we look back at the history of the creation of participatory processes in Europe, we can distinguish three main stages that have shaped the current movement towards establishing urban labs:

- The Scandinavian cooperative and participatory design movement from the 1960s and 1970s;
- The European social experiments with IT in the 1980s;
- The Digital City projects from the 1990s, where a digital city is understood as a place and its inhabitants implementing information and communication technologies [4].

The tradition of participatory projects dates to the 1960s and 1970s, when research projects on user participation in urban development were carried out in Sweden and Norway, among others [5]. As Strand and Freeman note, Scandinavian contributions to stakeholder theory over the past 50 years have played a much greater role in its development than is currently acknowledged. These contributions have included, among other things, a description of the term "stakeholder" or the first stakeholder map.

At first, participatory initiatives concerned designing solutions to make IT work better, resulting in the so-called *user-centred design approach* (UCD) coming to the fore. This

methodology treats the design of a product or service as a process in which the user's needs, requirements, and constraints are given special attention at each stage. Unique insights are identified, and innovative solutions are developed by asking about users' needs or observing them while using products or services. UCD and participatory design are still widely used in Scandinavian urban labs, which eagerly use proven methods to test prototypes of solutions and their usability in urban environments when designing solutions [6].

The second phase of the emergence of urban labs was the period of the creation of so-called *proto-living labs* in the 1980s, when social experiments with IT began across Europe. Social experiments originated in the field of psychology and refer to experiments taking place outside of laboratories. Researchers started to use social experiments as a test and implementation methodology in the context of the developing field of ICT in the 1980s. [4].

In the late 1990s, the digital city concept [7] was adopted in Europe as well as beyond. It refers to a range of digital initiatives undertaken by cities, especially those related to digital representations of the city, digitally related economic development and urban regeneration initiatives, and the provision of internet access for citizens [8]. The concept of a digital city was more ambitious than the previous two, as it combined citizens (users), policymakers (public organisations), and private organisations (businesses) on a large scale [9]. Thematically, the initiatives covered a wide variety of activities but were always related to city life. In terms of user involvement, it was seen as potentially innovative, and the technical infrastructure was only meant to be a trigger for this creativity.

Prior to the concept of the urban lab, there were individual mentions of the term "living laboratory" [10], but its conceptualization is attributed to Professor William Mitchell of the Massachusetts Institute of Technology (MIT), who from the mid-1990s onwards began to use it to refer to a specially designed laboratory in which the routine activities of experimental subjects and their interactions could be observed in the conditions of everyday domestic life [11]. The experiments conducted by Professor Mitchell were primarily "user"oriented and performed in a real environment. As a result of his research, he and his colleagues founded the first living lab research consortium, which was later reorganised into the MIT Media Lab.

Such labs, as well as other related user-centred innovation projects, emerged in Europe in the early 2000s, and Professor Mitchell was a member of expert groups in a number of them. These projects subsequently prompted the formation of the European Network of Living Labs [12]. Further development and evolution of the concept of urban labs occurred in European countries, notably in the Netherlands, Finland, Belgium, Austria, and Sweden. The combination of elements and experiences from previous decades is summarised in Table 1. For each previous process, it can be indicated whether a feature was already clearly present (+), somewhat present (+/−) or not present at all (−).

**Table 1.** Stages in the evolution of the urban (living) lab concept over the last decades.

|  | Cooperative Design, 1970s | Social Experiments, 1980s | Digital Cities, 1990s | Home Labs, 2000s | Urban (Living) Labs, 2010s |
|---|---|---|---|---|---|
| Active user involvement | + | +/− | − | − | + |
| Real-life setting | + | + | +/− | +/− | + |
| Multi-stakeholder | +/− | + | + | − | + |
| Multi-method approach | +/− | + | − | +/− | + |
| Co-creation | + | +/− | − | − | + |

Source: own study based on [4].

There is still no synthetic study that would systematise the names and definitions used to refer to functioning urban laboratories. Theoretical concepts and definitions relating to this instrument can be found in various foreign publications. The roots of the definitions of an urban lab and an urban (living) lab go back to the concept of the living lab, with which they share many common features. In 2010, it was noted that urban labs and living labs can be considered both a methodology and a space for users to initiate innovation processes by making use of the ideas, interests, and experiences of numerous stakeholder groups [13].

Based on the literature review, the authors identified 18 definitions of urban labs and urban living labs, treating these two types of labs equally. The most significant ones are summarised in Table 2, together with information on the most crucial groups of urban stakeholders involved in urban labs.

**Table 2.** Selected definitions of urban labs and urban living labs and their characteristics.

| Definition | Leaders and Others Involved | Author |
|---|---|---|
| Urban labs and living labs in general can be seen as both space and methodology for community participation with the purpose of initiating development processes that include ideas, interests, and experiences from multiple stakeholder groups. | Multiple stakeholder groups | [13] |
| Most of the proposed definitions describe urban labs as the loci in a given city where a group of people develop proposals and possibly experiment with and implement actions to address problems and challenges associated with that city. Urban labs can be established by local public administrations, which try to find new, more effective, and less resource-intensive modes of problem solving at the city level. | Local public administrations (leader)/ a group of persons | [14] |
| URB@EXP identifies urban labs as the same as living labs and city labs and defines them as an approach in which local governments engage in solving problems together with other stakeholders in urban development. | Local governments (leader)/ other stakeholders | [15] |
| The term refers to "the use of public city space—streets, buildings, or a designated neighbourhood—as an active laboratory where companies can evaluate and pilot pre-market products and services". | Companies | [16] |
| Urban labs are open innovation ecosystems, i.e., places, either promoted by companies or local institutions or spontaneously established by active citizens, where the current problems and challenges associated with a city are discussed and possibly innovative solutions are designed and implemented. | Active citizens, several heterogeneous actors | [1] |
| Urban living labs are being advanced as an explicit form of intervention capable of delivering sustainability goals for cities. ULL can be broadly conceived as forums 'for innovation, applied to the development of new products, systems, services, and processes, employing working methods to integrate people into the entire development process as users and co-creators, to explore, examine, experiment, test and evaluate new ideas, scenarios, processes, systems, concepts and creative solutions in complex and real contexts'. ULL scans can also be viewed as spaces designed for interactions between a context and a research process to test, develop, and/or apply social practices and/or technology to a building or infrastructure. | Users and co-creators | [17,18] |

**Table 2.** *Cont.*

| Definition | Leaders and Others Involved | Author |
|---|---|---|
| Urban living labs are emerging as a form of collective urban governance and experimentation to address sustainability challenges and opportunities created by urbanisation. ULLs have different goals; they are initiated by various actors; and they form different types of partnerships. There is no uniform ULL definition. Urban living labs constitute a form of experimental governance; whereby urban stakeholders develop and test new technologies, products, services, and ways of living to produce innovative solutions to the challenges of climate change, etc. | Various actors form different types of partnerships with urban stakeholders | [19] |
| ULL is a sort of system designed to experiment and co-create with the user the solutions that he or she will receive. Furthermore, it is a system in which end-users, together with various types of actors such as academics, companies, and public institutions, jointly research, design, and validate new and, above all, innovative products, services, and solutions to serve them. | End-users, including various types of actors such as academics, companies, and public institutions | [20] |
| Urban labs are a new form of governance, holding the potential to bring different actors together to work on sustainable solutions and to initiate mutual learning processes in which involved actors communicate and work at eye level, despite different social, economic, and political prerequisites, backgrounds, and resources. Urban labs are where the interaction between urban actors, stakeholders and researchers in experimental spaces is creating a new governance platform. | Urban actors, stakeholders, and researchers | [21] |

Source: own study.

On the basis of selected definitions of urban (living) labs presented in Table 2, it is possible to indicate their key features connected, among others, with urban stakeholders, city space, participation of inhabitants, taking initiatives by them, conducting urban experiments or testing ideas, and, as a result, implementing micro-innovations in cooperation with, e.g., companies, universities, or other organisations.

Emerging descriptions of the application of this instrument in real-life settings, e.g., in studies resulting from the European JPI Urban Europe project (2013, 2014) and in the comprehensive publication Guidelines for Urban Labs [22], also demonstrate the diverse approach to its functioning and definition.

Most of them indicate the activity of urban stakeholders to solve urban development issues and challenges, which can ultimately improve the quality of inhabitants' lives through the implementation of innovative solutions [1,14,17,19–21]. It is also worth noting that there is frequently a strong reference in the definitions to a physical space within a city [14]—often the seat of the urban lab—as a space designed for the interaction of various stakeholder groups [15,17] and which serves as a venue for public debate on the city's problems and possible solutions.

One of the main assumptions of urban labs is to establish strong collaboration between urban stakeholders. On the theoretical side, this is explained by the Quadruple Helix (QH) concept, which describes a model of cooperation to create innovation [23–25] between users, companies, scientific institutions, and urban (local) authorities [23]. In the literature on the subject, some authors focus on the role of different actors in co-creation in order to identify types of urban (living) labs depending on the main initiator or the most active participant [26]. Four different typologies are presented in Table 3.

**Table 3.** Typologies of urban labs distinguished according to the "initiator-leader" of the venture.

| | Authors of the Typology | | | |
| --- | --- | --- | --- | --- |
| | [23] | [27] | [28] | [3] |
| Typologies are distinguished according to the initiator-leader | Triple helix + users model | Utilizer-driven | University | Strategic |
| | Firm-centred model | Enabler-driven | Private corporation | Civic |
| | Public sector-centred model | Provider-driven | Multi-stakeholder partnership | Grassroots |
| | Citizen-centred model | User-driven | Community | |
| | | | Combination of various partners | |

Source: own study based on literature review.

In the case of the first of these, the division resulting from the concept of the quadruple helix (QH) is shown. On this basis, the authors of [23] distinguished four laboratory models. The first model of the triple helix focuses mainly on the development of commercial high-tech innovations based on the latest scientific knowledge, with users participating indirectly in the innovation process and decisions about actual user needs and their interpretation being made by experts. In the second model, a business representative plays a central role, and users share their knowledge and are treated as creators. Another model focuses on the development of public institutions and the services they provide. In this model, a public organisation is responsible for the innovation process and receives feedback from citizens together with partners (companies and other organisations). The last, citizen-centred model, focuses on citizens and the development of innovations relevant to inhabitants, who are central to the whole project process. New products, services, and ways of working are selected and developed by users. The role of companies, public authorities, and scientific institutions is to support citizens in innovation activities.

In the typology proposed by Leminen et al. [27], four types of urban laboratories are identified, depending on the "driving actors". In each, a different actor plays the most active role. *Utilizer-driven* labs are user-centred labs led by companies developing and testing products and services. The second type, *enabler-driven,* is oriented towards public sector activities and focused on local and regional development. The third type, provider-driven, is driven by development organisations to promote research and knowledge creation. The last type is *user-driven,* led by users and inhabitants focused on solving their problems with only indirect involvement of other stakeholders.

The typology by S. Marvin and J. Silver [28] presents five types of urban (living) labs, distinguished according to the establishing body, the entity that creates the urban lab. Type one, in which the university manages and directs the activities of the lab, funded by the academic sector and involving a number of partnerships with various entities, and coordinates and manages the R&D activities. In the case of a private corporation, the focus is on practical results that can be commercialised. In contrast, a multi-stakeholder partnership, often including universities, provides the opportunity to implement technology projects that require a large amount of funding. An urban lab can also be formed by a community (urban activists, academics, or students) whose main objective is to explore alternative ways of neighbourhood development. A combination of various partners are labs that are a mix of several types, where different actors work together in a given space. The last two cases, according to the authors' research [28], are the least frequent in urban settings. However, the authors of this paper observe a growing interest in labs created and managed by residents.

The last typology distinguishes three types of urban labs in terms of the type of leadership, the subject matter undertaken, and the scale of operation [3]. Type one *strategic* is led by the city or large private companies. It uses the city area to implement projects involving different partners. Type two *civic* is led by a university or the city and focuses on sustainable urban development. Type three *grassroots* is led by inhabitants (urban activists) focusing on the quality of life and economy, often implementing micro-projects with a limited budget.

Each of the cited typologies clearly highlights the role of users (inhabitants) in creating and managing urban lab activities, as well as the highly placed potential of universities [28], the city, and private companies [3]. In these cases, citizens are treated as partners in the activities undertaken and can bring great added value, especially when implementing smaller-scale tasks and projects. Moreover, it is worth noting the evaluation of the typology over the years as well as the scale of activities within the urban space—from micro-projects in the last type to multidimensional ones involving different urban stakeholders in the space of the whole city in the first example.

*3.2. Results: Analysis of the Results of a Survey on the Functioning of Urban (Living) Labs in the World*

In order to collect detailed information on the functioning of urban (living) labs around the world, a qualitative and quantitative questionnaire was developed, containing 35 closed and open questions. The questionnaire form was made available online and sent by e-mail to identify and select 39 urban labs operating in various countries around the world. As a result, 24 completed questionnaires were accepted for final analysis, representing 61.5% of all urban labs invited to the study (Table 4). The following section presents selected survey results and conclusions.

**Table 4.** Urban (living) labs participating in the survey and the main purpose of their creation.

| No. | Country | City | Name of Urban (Living) Lab (Year of Creation) | The Main Purpose of Creation |
|---|---|---|---|---|
| 1. | Canada | Calgary | EVDS Urban Lab (2000) | Conducting research, education, and assistance with issues related to urban design, developing and applying research and analysis methodologies to facilitate professional experience for students. |
| 2. | Estonia | Tartu | Mobility Lab (2004) | A deeper understanding of spatial mobility using ICT tools and location data. This is a working group at the university. |
| 3. | UK | London | UCL Urban Laboratory (2005) | Fostering and promoting the dialogue between social science researchers and the creation of environmental disciplines at University College London. |
| 4. | Estonia | Tallin | MTÜ Linnalabor-Estonian Urban Lab (2006) | Introducing changes in urban planning with the participation of inhabitants. |
| 5. | Spain | Cornellà de Llobregat | Citilab (2007) | Developing the knowledge society. |
| 6. | Mexico | Querétaro | Laboratorio Urbano Queretaro (2008) | Supporting urban research based primarily on three thematic groups: urban arrangement, civic participation, and mobility. |
| 7. | USA | Nashville | Urban Green Lab (2009) | Offering an insight into and understanding of the theory of sustainable living. |
| 8. | Scotland | Glasgow | Glasgow Urban Lab (2009) | Conducting research on the city of the future. |

**Table 4.** *Cont.*

| No. | Country | City | Name of Urban (Living) Lab (Year of Creation) | The Main Purpose of Creation |
|---|---|---|---|---|
| 9. | Sweden | Malmö | STPLN (2011) | Facilitating and developing creative grassroots initiatives (individual and group), especially in the cultural and creative spheres, the sharing economy, and 'zero waste' activities. |
| 10. | Armenia | Yerevan | Urbanlab Socio-Cultural Foundation (2011) | Sharing experiences, promoting public participation, and broadly defining sustainable development. |
| 11. | USA | New York | The GovLab (2012) | Improving the quality of inhabitants' lives as a result of more efficient city management by using new technologies to combine two potentials: city data and inhabitants themselves. |
| 12. | Uganda | Kampala | Urban Action Lab (2012) | Identifying, prioritising, and conducting research on relevant urban policy issues. |
| 13. | Belgium | Antwerp | Antwerp Citylab2050 (2012) | Co-creating projects and experiments related to various topics that, in the long run, will contribute to the sustainable development of the city. |
| 14. | Italy | Cesena and Bologna | Smart City Lab (2012) | Conducting research related to urban technological innovation. |
| 15. | The Netherlands | Groningen | Urban Gro Lab (2013) | Building stronger cooperation between the university and the city. Conducting research (by students and researchers) and attempting to answer questions related to the functioning of the city and its future, in the form of studies, experiments, and other projects. |
| 16. | Germany | Nuremberg | Urban Lab Nürnberg (2014) | Co-creative transformation/co-productive transformation: increasing cooperation with inhabitants, NGO representatives, and city administration. |
| 17. | Spain | Catalonia—various cities | SmartLAB (2014) | Activating an innovative ICT sector. |
| 18. | Ukraine | Kiev | UrbanLabKyiv (2014) | Investigating the transformation of post-Soviet cities, conducting urban studies, coordinating urban workshops, developing culturally "driven" public spaces, urban anthropology. |
| 19. | Colombia, Germany | Medellin, Berlin | Urban Lab Medellín\|Berlin (2016) | Exchanging knowledge between the informal neighbourhoods of Moravia and Medellín and experts and students from Berlin in order to support the community and involve them in the transformation of their neighbourhood. |
| 20. | Canada | Hamilton | CoLab (2016) | Building collaboration between the Hamilton community and McMaster University stakeholders in the field of common research interests and goals. |
| 21. | Spain | Barcelona | BCNUEJ (2016) | Conducting research and taking actions towards sustainable urban development. |
| 22. | Austria | Graz | Mobility Lab Graz (2017) | Providing an innovative environment in the region of Graz to foster innovation in the mobility sector with the overall objective of reducing individual transport. |

**Table 4.** *Cont.*

| No. | Country | City | Name of Urban (Living) Lab (Year of Creation) | The Main Purpose of Creation |
|---|---|---|---|---|
| 23. | Mexico | Merida | Laboratorio Urbano del Mayab (2017) | Cooperating in the development of urban policies that favour the environment and the inhabitants' health. |
| 24. | Austria | Vienna | aspern.mobil LAB (2017) | Testing and implementing innovations in a real setting. |

Source: my own study based on the survey.

Among the laboratories surveyed, those from European countries clearly prevail, accounting for 62% of the total sample, followed by 25% from North America (Canada, Mexico, and the USA). The remaining three represent the African continent—Uganda (Kampala), Asia—Armenia (Yerevan), and South America—Colombia (Medellin). Importantly, the laboratories analysed were established after 2000, indicating that they are relatively new instruments in all countries, the oldest being Canada's EVDS Urban Lab, established in 2000 in Calgary.

When analysing the most significant purposes of establishing urban labs (Table 4), it should be noted that they are very diverse and can be divided into two groups: technological and social. Among the technological objectives of creating the labs, the following were mentioned: activating an innovative ICT sector; conducting research related to urban technological innovation; opening and sharing urban data, which are clearly dominant in the case of the Catalan SmartLAB, the Smart City Lab from Cesena and Bologna, as well as the Mobility Lab from Tartu and The GovLab from New York. The purposes for the creation of the remaining urban labs mainly included those of a social nature: research into the functioning of the city, promoting social participation, building cooperation between inhabitants, and developing grassroots local initiatives. Testing and implementing innovations in the real world is one of the main tasks of urban labs.

For each of the urban labs surveyed, the founding body eventually became its managing body. The exceptions here are Citilab, Urban Gro Lab, and aspern.mobil LAB. The managing entities in the studied urban labs were grouped into elements according to the quadruple helix concept [23]:

1. Scientific institutions 50%;
2. Inhabitants 29%;
3. City authorities 13%;
4. Mixed 8%;
5. Business 0%.

The highest percentage are urban labs managed by "scientific institutions" (50%). Almost 1/3 of the urban labs surveyed were founded by inhabitants (29%), with the managing entities in these situations being NGOs or the local community directly. Only 13% of local authorities are managing bodies. Mixed cases were also included (8%), where the lab is jointly managed by at least two representatives of the above-mentioned stakeholder groups.

The entities involved in urban labs are grouped according to the elements of the quadruple helix [23].

The local government and scientific institutions are most often involved in the functioning of urban labs (30% each). Slightly less engaged are NGOs and/or local activists (23%) and entrepreneurs (17%) (Figure 1). When analysing the actors involved in the activities of individual urban labs, it is worth noting that "local activists" are always in partnership with the "local government". Some urban labs engage representatives of all stakeholder groups, others only a part of them. From 3 to 5 entities are usually involved in urban labs.

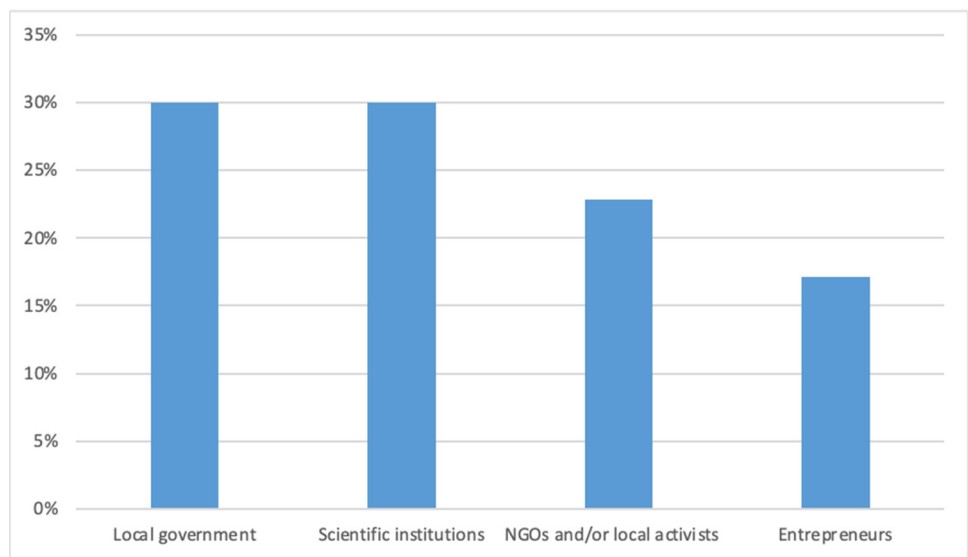

**Figure 1.** Share of various urban stakeholder groups involved in urban labs (according to the quadruple helix concept). Source: my own study based on the survey.

On the basis of the surveys conducted, it is noted that not every urban (living) lab has a physical location in the city space. However, in the question concerning having a seat, as many as 79% of the laboratories surveyed answered in the affirmative. In the case of some urban labs, they do not have separate, independent premises but, for example, use a dedicated workstation at the university or in city administration buildings. The question about physical space was answered in the negative by 21% of the urban labs. Their employees often work together without a permanent meeting place or have only a dedicated smaller space in the institution with which they cooperate or that manages the lab (e.g., universities).

The majority of the urban labs analysed (54%) have between 1 and 5 employees in their units (Table 5). Eight percent of the labs employ between 16 and 20 people, but in no case did the number of people exceed 20. The number 0 refers to a dedicated workstation at the university or in public administration. There is no clear correlation between the number of people employed and the location of the urban lab or the subject of its activities.

**Table 5.** Number of urban lab employees.

| Number of Employees | Percentage of Urban Labs Surveyed [%] |
|---|---|
| 0 people | 13 |
| 1–5 people | 54 |
| 6–10 people | 17 |
| 11–15 people | 8 |
| 16–20 people | 8 |

Source: my own study based on the survey.

Projects are realised by urban labs at various spatial scales, from local to international, and these correspond to the relevant networks (Table 6).

**Table 6.** The spatial scale of implemented projects and network affiliation.

| No. | Name of the Urban Lab | The Spatial Scale of the Project | Cooperation Network |
|---|---|---|---|
| 1 | EVDS Urban Lab | Local, regional | The Urban Alliance (City of Calgary and University of Calgary) |
| 2 | Mobility Lab, University of Tartu | Local, regional, and international | COST networks |
| 3 | UCL Urban Laboratory | Local, regional, and international | Urban Lab+ |
| 4 | MTÜ Linnalabor-Estonian Urban Lab | Local, regional | X |
| 5 | Citilab | Local, regional, and international | European Network of Living Lab (ENoLL) |
| 6 | Laboratorio Urbano Queretaro | Local, regional | X |
| 7 | Urban Green Lab | Local, regional | X |
| 8 | Glasgow Urban Lab | Regional, international | UNECE Academy of Urbanism |
| 9 | STPLN | Regional | Anna Lindh Foundation, European Creative Hubs Network |
| 10 | Urbanlab Socio-Cultural Foundation | Local, regional, and international | Docomomo International |
| 11 | The GovLab | Local, regional, and international | X |
| 12 | Urban Action Lab | Local, regional | Urban Climate Change Research Network |
| 13 | Antwerp Citylab2050 | Local | X |
| 14 | Smart City Lab | Regional, international | X |
| 15 | Urban Gro Lab | Local | X |
| 16 | Urban Lab Nürnberg | Local, regional | Verbund offener Werkstätten, Urbane Liga |
| 17 | SmartLAB | Local | Technological clusters |
| 18 | UrbanLabKyiv | Regional, international | X |
| 19 | Urban Lab Medellín | Berlin | Local | ARCH+ Association |
| 20 | Community Campus CoLaboratory | Local | X |
| 21 | BCNUEJ | International | ICLEI |
| 22 | Mobility Lab Graz | Regional | Austrian urban mobility labs |
| 23 | Laboratorio Urbano del Mayab | Local, regional | Instituto de ciudades en movimiento y red de laboratorios urbanos |
| 24 | aspern.mobil LAB | Local, regional | X |

Source: my own study based on the survey. The highest percentage of the projects carried out in the examined urban labs concern those on a local scale (40%), followed by activities on a regional scale (38%), and in the case of 22%, on an international scale. Most of the urban labs researched (14) belong to various networks and associations. These are the majority of organisations operating on a national scale. In some cases, they are also international networks, such as ENoLL (the European Network of Living Labs).

Urban lab project funding is very diverse. Its type depends to a large extent on the country in which the lab operates and its policy of supporting similar instruments. The aim of the lab (main theme and specialisation) and the management are also relevant. At the same time, the source of financing largely determines the way in which the urban lab operates. Public funding, for example, on the one hand offers the possibility of addressing a wider audience, as it usually includes the obligation to disseminate activities and to

support the introduction of innovative practices [15], while on the other hand, it can be somewhat limiting due to complicated administrative procedures [2].

With very few exceptions, funding comes from more than one source, mainly from public, municipal, and non-governmental sources, research grants, as well as private sources (funding from commercial companies) and self-funding. Funding in the form of grants from the public sector predominates. The four urban labs are largely covered by ministerial grants. Private funding is only partially present as one of several sources of funding (Table 7).

**Table 7.** Sources of project financing for the surveyed urban labs.

| No. | Name of the Urban Lab | Sources of Project Financing |
|---|---|---|
| 1 | EVDS Urban Lab | University grants, Government of Canada grants for research, City of Calgary and neighbourhood associations, some funds from the private sector. |
| 2 | Mobility Lab, University of Tartu | Estonian Research Council; Horizon 2020 Programme, ESPON Programme, Commission of the European Communities. |
| 3 | UCL Urban Laboratory | Internal university grants (e.g., Global Engagement, Public Engagement, Grand Challenges) and external grants (e.g., ESRC, AHRC, HERA). |
| 4 | MTÜ Linnalabor-Estonian Urban Lab | Local governments and local resources such as the Cultural Endowment of Estonia. We are looking for separate funding for each project we want to run; we do not have regular funding. |
| 5 | Citilab | Mainly a public authority. |
| 6 | Laboratorio Urbano Queretaro | The projects are externally funded by the Mexican National Council for Science and Technology, the British Academy, and IBM Research Group. |
| 7 | Urban Green Lab | All, but primarily government subsidies. |
| 8 | Glasgow Urban Lab | UK Research Councils, public and private sectors. |
| 9 | STPLN | Basic funding (including premises) comes from the municipality, national and international funding (including the EU), and 25% self-financing through workshops, rent, fees, etc. |
| 10 | Urbanlab Socio-Cultural Foundation | Some projects are self-financed; others are implemented through various grants and with the support of local and international organisations. |
| 11 | The GovLab | Foundations, government partners, international organisation partners, and NGO partners. |
| 12 | Urban Action Lab | Foreign Funding Agencies. |
| 13 | Antwerp Citylab2050 | Local funding, European funding, and regional funding. |
| 14 | Smart City Lab | Private and public funding |
| 15 | Urban Gro Lab | Financed by the municipality |
| 16 | Urban Lab Nürnberg | Oftentimes, our major projects are financed by ministerial funds (BBSR and NSP). Other financial sources include local companies, foundations, and income from our own projects. |
| 17 | SmartLAB | It is free testing of solutions; companies fund their part, and cities assist them in testing. |
| 18 | UrbanLabKyiv | Grants, private investors, and commercial research projects. |
| 19 | Urban Lab Medellín\|Berlin | Foundations, German Academic Exchange Service (DAAD), German Embassy in Bogotá, Municipality of Medellín, private companies. |
| 20 | Community Campus CoLaboratory | Grants from the Ontario Trillium Foundation. Grants from the Social Sciences and Humanities Research Council. |
| 21 | BCNUEJ | EU projects (ERC and Horizon 2020), Spanish funds, and Catalan funds. |
| 22 | Mobility Lab Graz | Federal Ministry of Transport, Innovation, and Technology; City of Graz; and Land Steiermark. |
| 23 | Laboratorio Urbano del Mayab | Various. |
| 24 | aspern.mobil LAB | Ministry of Transport, Innovation, and Technology. |

Source: my own study based on the survey.

Analysing the results concerning the thematic areas within which urban labs operate, the following fields of action can be distinguished: technological innovations, social innovations, spatial management, data opening, and innovation incubators.

Most of the surveyed labs mentioned in their activities the implementation of projects, including research and educational ones, of a pilot or online nature. Opening and sharing public data came in second, while an innovation incubator operates in half of the surveyed labs (Table 8).

**Table 8.** Scopes of action of urban labs, subject matter of undertaken activities, and exemplary projects.

| Scopes of Action | Thematic Areas | Main Topic of the Projects |
|---|---|---|
| Technological innovations | - Mobility;<br>- Sustainable development;<br>- New technologies;<br>- Development of innovation policies. | - Urban traffic management;<br>- Urban logistics;<br>- Autonomous driving;<br>- IoT;<br>- Traffic management 2.0;<br>- Road safety ("vision zero" strategy). |
| Social innovations | - Interdisciplinary urban research;<br>- Participation;<br>- Inclusion of communities. | - Healthy ageing;<br>- Campus areas as laboratories for participatory urban design;<br>- Crowdlaw. |
| Spatial management | - Public space;<br>- Urban planning;<br>- Urban forms;<br>- Green gentrification. | - Adaptation;<br>- Urban resilience;<br>- African cities;<br>- Peculiar infrastructure;<br>- Feminist cities;<br>- Critical urban heritage;<br>- Construction of small urban architecture. |
| Data opening | - Release of government and corporate data. | - Mobile positioning data for tourism statistics;<br>- Sharing data from the city and mapping;<br>- Data on inhabitants' movement around the city;<br>- Collecting information from inhabitants about their ideas for the coastal area;<br>- Weather data, spatial data;<br>- Building small sensors during workshops;<br>- Data on pollution and climate change;<br>- Georeferenced road safety information system for decision making. |
| Innovation incubator | - Supporting startup projects;<br>- Sharing space;<br>- Mentoring and supporting creative projects. | - Advising young lecturers and assisting them in conducting research;<br>- Workshops to facilitate the identification of the theory of change. |
| Other | - Zero waste;<br>- Urban culture policy;<br>- Alternative and non-formal education activities;<br>- Tourism;<br>- Art and technology;<br>- Co-production. | |

Source: my own study based on the survey.

The formula of the urban café, consisting of, among other things, the organisation of meetings with the inhabitants in the city space, during which ideas are presented and examples of solutions are discussed, is applied only in two urban labs. These meetings are held irregularly in city cafés, but in no case is there a permanent, physical location within the city space.

In terms of both the thematic areas within which activities are undertaken in the surveyed urban labs and the subject matter of the specific projects they run, those of a social nature dominate and definitely stand out in number in relation to the technological ones.

Part of the questions in the questionnaire concerned the identification of the most important problems and barriers encountered during the operation of the urban lab. They are listed in six groups in Table 9.

**Table 9.** Groups of problems and barriers identified in the activities of urban labs.

| Groups of Problems and Barriers | Problems | Barriers |
|---|---|---|
| **Finances** | - Obtaining financial support for the beginning;<br>- Obtaining funding for equipment necessary for work;<br>- Financing of non-commercial activities. | - Difficulty in finding a sustainable funding stream;<br>- Lack of regular funding results in irregular staff work;<br>- Funding opportunities requiring the need to start new projects instead of building on previous activities;<br>- High financial dependence on new projects;<br>- Obtaining projects that can be implemented quickly;<br>- Short-term capacities. |
| **Staff** |  | - Shortage of people with sufficient competence to implement projects;<br>- Uncertain guidance;<br>- Learning to operate systemically;<br>- Only one person running the lab—therefore it has never become a platform;<br>- Maintaining team cohesion;<br>- Communication problems;<br>- Lack of an office. |
| **Communication** | - Reception of information about what we do and how we uniquely solve problems;<br>- Building credibility and recognition regarding a pluralistic approach;<br>- Difficulty in explaining what we do. | - Problems in communicating the activities of the lab, directing, and filtering information;<br>- Promotion of the lab in research, economy, and society;<br>- Making residents aware of the advantages of the urban lab and the services it offers. |
| **Concept** | - A vague concept that is difficult to grasp;<br>- Starting from scratch. | - Development of a lab model for the inhabitant. |
| **Administration** | - Breakdown of previous approach to organisation—moving from hierarchical to participatory;<br>- Limited institutional support;<br>- Bureaucracy. | - Obtaining acceptance by the city administration;<br>- Maintaining an interdisciplinary identity at the university;<br>- Political reluctance. |
| **Cooperation between various stakeholder groups** | - Coordination of various stakeholders;<br>- Management of creative people;<br>- Activation of users. |  |

Source: my own study based on the survey.

The most frequently mentioned common group of problems and barriers were financial issues, such as the acquisition and regularity of funding to commence activities. Then communication related to the inhabitants' proper reception and understanding of the activities and appreciation of the proposed solutions. A common problem also includes the development of the model as well as administrative issues.

## 4. The Model and Discussion of Its Elements

As a result of the work carried out in the Institute of Urban and Regional Development to elaborate the concept of an urban lab adapted to the conditions of the countries of Central and Eastern Europe, which was a response to the lack of this type of instrument in Polish cities, as well as on the basis of literature analysis and questionnaire surveys of existing urban labs in the world, the author's model of the functioning of an urban lab was created. The main components of the model are the urban lab's stakeholders, the thematic areas and methods of its operation (the project process), its evaluation, and the good practices developed (Figure 2).

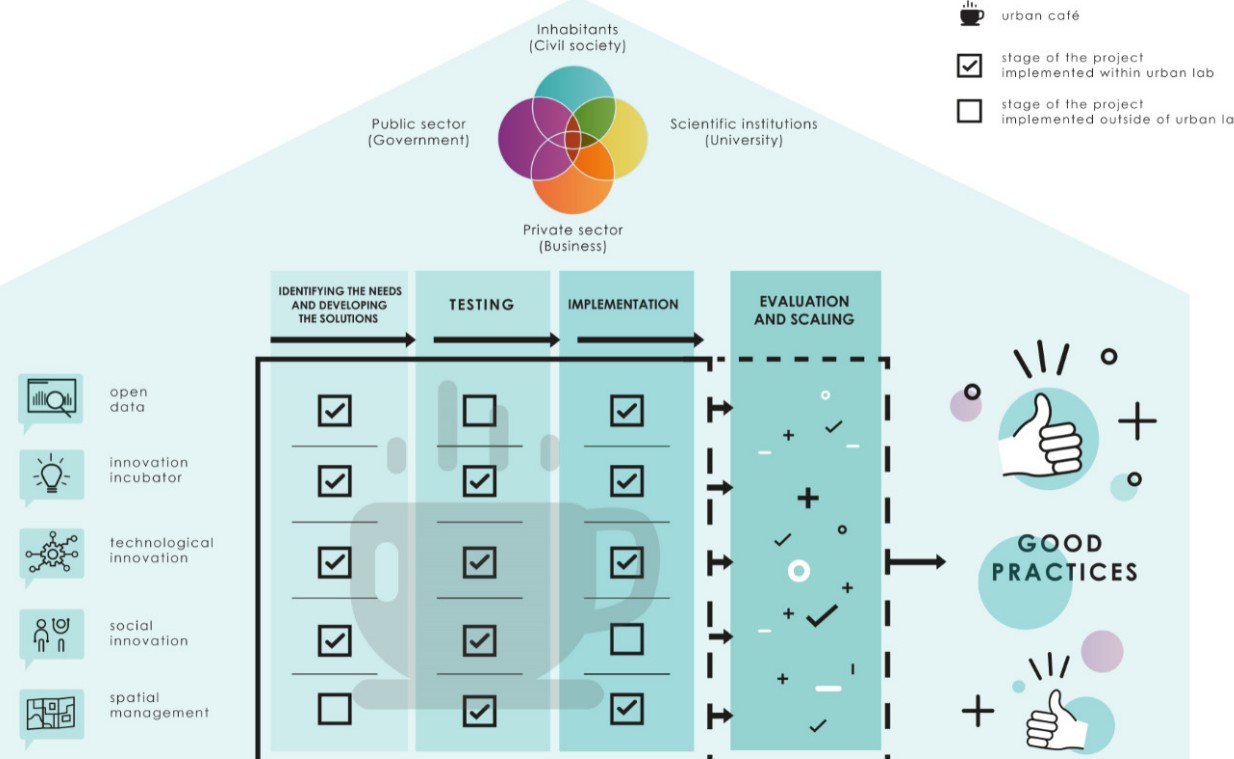

**Figure 2.** A model view of an urban lab adapted to the conditions of Central and Eastern European countries. Source: my own study.

In the developed model, the different stakeholders of the urban lab (inhabitants, public sector, private sector, and scientific institutions) correspond to the elements of the quadruple helix, who cooperate to produce innovative solutions [23]. Their roles can be very diverse, depending on the collaboration. Local stakeholders do not often stimulate innovation and entrepreneurship [29]. They are more responsible for creative activities in the city, which need to be translated into a structured process of social and technological innovation. In this way, initiatives coming directly from the inhabitants have a chance to scale up, be tested in real settings, and be implemented on the basis of adequate resources and with the formal leadership of one of the partners. The shape of the themes of projects implemented in urban labs and their management are influenced by the relationships between individual stakeholders and the roles they take on.

Within the scope of the laboratory's activities, the issues of open data, urban innovation incubators, technological innovation, social innovation, and urban space management are addressed, which have been included in the model on the basis of the presented surveys on the experiences of various laboratories from all over the world. The results of these surveys and the analysis of the literature on the subject show that in many labs, only selected topics are dealt with. The boundaries between the ranges of activities performed are not strict,

and their topics may overlap, but certainly the scopes mentioned should complement each other.

Four stages of project implementation are characteristic for the different fields of action of urban labs: research of users' (inhabitants) needs and development of solutions; testing of the developed solutions; implementation in real settings; as well as their evaluation and possible scaling [18,30]. Testing can be piloted and limited in scale. Urban labs may implement all the steps for the activities undertaken or only select ones, and some of them may take place outside the lab. Each activity should be concluded with an assessment of its usefulness in the form of evaluation and applicability on a different scale. For some solutions, replicability is not necessary, but scalability for implementation in other settings is essential.

An urban lab is also intended to be a place for inspiring meetings, a space for the exchange of modern urban ideas, and a place where numerous thematic events will be organised with invited experts in their field, for the inhabitants, and with their participation. A conversation over a cup of coffee was assumed to be a pretext for urban discussion, which is why the so-called urban café plays a major role in the presented model. The urban café is a physical part of the urban lab, where thoughts and ideas about the future of the city are exchanged. This creative space additionally influences the activation of the local community and its inclusion in the decision-making processes of the city. The urban café formula can be used for projects dealing with a wide range of issues at various stages of their implementation and can take the form of discussions, debates, workshops, or other events. Urban cafés are one of the means of supporting the achievement of the goals set in urban labs. That is why a "cup of coffee" appears on the model in the background of all project stages. Solutions successfully tested in a city should be scaled up and collected in a catalogue of good practices for use in other cities.

## 5. Conclusions

Various definitions or typologies of urban (living) labs presented in the paper, as well as different approaches to their activities in numerous countries around the world, show how complex and heterogeneous a tool they are. In spite of the noticeable differences, their overriding aim is invariably to improve the quality of life of city dwellers, taking into account the interests of each group of urban game actors, who can act as providers and/or recipients of the developed solutions.

As part of the work on the creation of a model and concept of the urban lab adapted to the conditions of the countries of Central and Eastern Europe, a definition of the urban lab that is as precise and wide as possible in context was developed: it is an instrument-an organisation and physical space (office and/or part of the city chosen for testing selected solutions) of cooperation between city authorities and inhabitants (including in particular those represented by non-governmental organisations, urban activists and or social activists), enterprises (from local micro-enterprises to global corporations) and scientific entities (universities, scientific and research units, experts), aimed at improving the quality of life of inhabitants through innovative solutions to identified problems (initiating, testing, implementing and evaluating projects) and generating additional value using city resources. In this definition, the main elements appearing in the definitions of urban (living) labs, living labs, city labs, and smart labs from the literature on the subject have been included in order to capture the essence of the functioning of an urban lab as completely as possible.

In order to maximise the benefits of their engagement with an urban lab, it is crucial to communicate its purpose, structure, and activities in a transparent way [31]. The growing interest in urban labs is reflected in an increasing number of publications on the subject of their operation, particularly those containing case studies from projects carried out in various cities around the world. There are also articles that pay attention to the relationship between living labs and citizen sciences and their impact on participatory processes in the city [32]. Engez et al. (2021) [33] analyse living labs as ecosystems, enabling economic value flow, material flow, and knowledge flow, and pursuing the shared goal of improved

environmental sustainability. There is plenty of research, too, on how urban (living) labs (ULLs) can become pathways for a sustainable transition towards innovative urban systems from a circular economy perspective, as well as analysing them as instruments with a real impact on sustainability through real-world experimentation, leading to the implementation of innovations and promoting change in urban ecosystems [34].

This draws the attention of other cities considering their implementation and, at the same time, provides a good practice base for institutions engaging in this type of cooperation. The development of the third generation of the smart city concept [35], for which social, educational, inclusive, or ecological issues are above all characteristics—in addition to urban projects with technological tools—means that in this case urban innovations are implemented in an open, continuous process in which inhabitants play a key role. This increases the popularity of the urban lab, a solution in which the role of the urban space user—i.e., an inhabitant—in co-managing the city is clearly defined and the inhabitants themselves, thanks to their activity, are increasingly appreciated for their contribution to urban development. Examples of very active urban labs in different countries around the world pursuing similar goals include Urban Lab Nürnberg, aspern.mobil LAB from Vienna, Urban Lab Medellín | Berlin, or STPLN from Malmö [36].

**Author Contributions:** Conceptualization B.P., M.B. and W.J.; methodology B.P., M.B. and W.J.; validation B.P., M.B. and W.J.; formal analysis B.P. and M.B.; investigation B.P., M.B. and W.J.; resources B.P., M.B. and K.N.; data curation M.B. and K.N.; writing—original draft preparation B.P., M.B. and K.N.; writing—review and editing B.P. and M.B.; visualisation M.B. and K.N.; supervision B.P. and W.J.; project administration B.P. and M.B. All authors have read and agreed to the published version of the manuscript.

**Funding:** This paper was written within the project entitled "Urban Lab as a pilot tool to improve the quality of life of city residents in line with the smart city concept" which was carried out at the Institute of Urban and Regional Development in cooperation with the Ministry of Development Funds and Regional Policy and co-financed by the European Union under the Operational Programme (OP) Technical Assistance (TA) 2014–2020.

**Institutional Review Board Statement:** Not applicable.

**Informed Consent Statement:** Informed consent was obtained from all subjects involved in the study.

**Data Availability Statement:** ResearchGate: https://www.researchgate.net/ (accessed on 2 February 2023); Infona: https://www.infona.pl/ (accessed on 20 December 2022); Google Scholar: https://scholar.google.com/ (accessed on 2 February 2023).

**Conflicts of Interest:** The authors declare no conflict of interest.

### Appendix A. List of Questions Included in the Survey Sent to 39 Urban Labs (The Survey Was Conducted in the Period November 2018–April 2019)

1. Name of your urban lab
2. City of location
3. Date of creation
4. Who founded your urban lab?
5. Who mainly manages the urban lab?
6. What was the main goal of creating the urban lab?
7. When creating your urban lab, were you inspired by other examples from different cities (if yes, please indicate this urban lab/s below)?
8. Do you have any physical offices?
9. If yes, please explain where it is located.
10. What is the main thematic area of the urban lab?
11. What are the main topics of your projects, please indicate the most important?
12. What is the scale of the projects?

[local/regional/international/other]

13. From what sources are projects financed?
14. Are the following entities involved in the urban lab activities?

[local government/universities/science/research institutions/entrepreneurs/NGOs/ local activists]

15. What is the local government responsible for? Please explain.
16. What are universities/science/research institutions responsible for? Please explain.
17. What are entrepreneurs responsible for? Please explain.
18. What are the NGOs responsible for? Please explain.
19. What are the local activists responsible for? Please explain.
20. If there are other entities, what are they responsible for? Please explain.
21. For the needs of our project, we have developed four groups of tasks within our future urban lab activity. Which of these tasks occur at your urban lab?

[Sharing and using urban data/Initiating, testing, and implementing projects/Managing an urban café/Creating and coordinating incubator activities]

22. If occurring, please describe the tasks of sharing and using urban data:
23. If occurring, please describe the tasks of initiating, testing, and implementing projects:
24. If occurring, please describe the tasks of managing an urban café:
25. If occurring, please describe the tasks of creating and coordinating incubator activities:
26. If any other activities are occurring, please describe:
27. How many people are permanently employed at your urban lab?
28. Is the urban lab associated with any network?
29. If yes, what is the name of the association?
30. Do you monitor the functioning of the urban lab?
31. If yes, what are the ways to do it?
32. Do you prepare any reports summarising the urban lab activity?
33. If yes, please write more details about the reports, or if you published any document online, please paste a link to the website.
34. What kind of problems did you meet during the process of creating the urban lab and at the beginning of its existence?
35. What are the biggest barriers or problems that you face during your urban lab activity?

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
