# Peer review of "Exploring Urban (Living) Labs: A Model Tailored for Central and Eastern Europe’s Context"

_sustainability, doi:10.3390/su151612556_

Round 1

Reviewer 1 Report

Your abstract seems well organized, but the length is not enough and it is suggested to expand it, and the results part in the abstract lacks data, not recommended to use text only.

General comment on the Introduction section: my main suggestion is to expand the introduction that is a bit too short and to make a deeper analysis of the most recent literature. 

The research method is too simple and needs to be described in detail. Including research conditions, processes, precautions, etc.

The format of the paper needs to be modified according to the template of the journal. 

Your discussion part is very well written.

Conclusions: Further focus on your results/findings.

I have no strong plagiarism checker and you should do that.

Reviewer 2 Report

Dear authors,

the paper presents interesting and useful research on a collective initiative (living labs) aiming the improvements of societies and development. In order to improve the focus and the impact of the research carried on, there are some aspects that deserve your attention:

1. What is the linkage with the issue of sustainability? It is important to clarify this link considering that this is a journal on sustainability.

2. It is not clear the relation between the research done in 17 countries and its application in central and western Europe; what is exactly the regional/country purpose?

3. What were the criteria used in the selection of the cases addressed in different countries?

4. What is the difference between living labs and other iniciatives involving multiactors and, sometimes, multiscale scopes, such as local development associations, cooperatives, etc.

5. What are the main impacts of this initiatives in the sustainability of the territories where they operate (linked with 1)?

Reviewer 3 Report

Firstly, I would like to congratulate the authors for writing the article entitled: Attempt at Summarising Studies on Urban (Living) Labs. A proposal of an Urban Lab model adapted to the conditions of Central and Eastern Europe. The paper looks well-structured having potential to contribute to the literature on the topic. However, to enhance the academic quality, several revisions are necessary prior to its publication.

-          The title (but of course this is now to late to change it) does not give strength to the article – even if the content is good, mostly because it starts with “Attempt” and also includes “proposal”. It might send the reader to search for weakness – “is the attempt successful or not?!”, “is the proposal worth considering?”. Moreover, it does not sound very English eighter. If possible, I would suggest reshaping it.

-          While the title refers to CEE countries, the conclusions starts with: different approaches to their activities in numerous countries of the world… this lack of convergence should be avoided in a scientific article – or explained in order to be properly understood by the reader;

-          As an advice, better not use in the keywords list, words that are to be found in the title – both of those fields are indexed in data-bases so better use something that are more helpful for other researchers to find (and cite) the article;

-          The abstract lacks providing relevant info about the article. It should say more about the studied countries, a bit about the methodology and results in order to gain attention to potential readers. Consider rebuilt it from scratch an add more flavor;

-          Introduction also needs to be developed furthermore and should be followed by a large Literature review section (that might contain a substantial part of 3.1. Theory: The origin of […] section) in order to fit the title – […] Summarising Studies […] and the aim of the study, which is, according to the article: “to synthesise existing knowledge and research […]”, and the following statement “For the purpose of this paper, a systematic literature review was carried out”;

-          Also, for the Literature review section there are plenty of scientific articles all around MDPI journals that could inspire and worth being cited. In the same time, I would advise browsing the articles of the Smart Cities and Regional Development (SCRD) Open Access Publishing Journals (www.scrd.eu) as reference – polish authors and others from CEE countries published there on the same topic;

-          Materials and Methods section should also be extended by providing some relevant items that were present in the survey – if not all, and the tools (either mathematical, statistical or software) that were used. The whole survey would be an important piece of the article as an appendix;

-          It is also important to know on what basis the countries were chosen especially since the title indicates CEE countries;

-          The timeframe is a bit old, 2018-2019. Maybe a refresh worth being considered – lot of things happened meanwhile, things that were able to bust up the way citizens are interacting with municipalities;

-          The beginning of the section 3.2 Results: Results: Analysis of […] fits better to Materials and Methods section. I reiterate the need of an Appendix with the survey items;

-          The statement on line 211-212 is too strong. I understand that it refers to the sample, however, the sample itself is not statistically relevant (or the relevance is not quite clear since we have no idea about how other countries, which were not part of the study, are tackling the issue, nor about those who didn’t answer to the survey);

-          From Table 6 it jumps to Table 8 – must be a mistake. However, Table 7 is mentioned in text… that needs to be corrected;

-          I would advice not using bullets inside Table 9. Groups of problems […];

-          Section 4 Discussion has one subsection 4.1. The model […]; I see no reason to have one subsection only. Maybe the authors are willing to divide it in two or more subsections;

-          Conclusions are consistent with the study (and the results);

-          References are old with very few newer then 2020. I would suggest paying attention to that aspect too, which correlated with advices on the literature review section, could result in a more interesting article;

Once again congratulation to the authors.

Round 2

Reviewer 2 Report

Dear authors,

The paper was improved. As already mentioned in previous revision, this is an important topic. It is very important that it presents results from research-action. It is also perfectly clear, for me and other academics, that publication finishes a path, meaning that something important, and impactful, should be shared. 

Author Response

Thank you very much for feedback and all comments.

With regards

Reviewer 3 Report

Congratulations for your efforts. 

Author Response

Thank you very much for your feedback and all comments.
With regards